# Designing and Planning of Studies of Driver Behavior at Pedestrian Crossings Using Whole-Vehicle Simulators

Rafał Burdzik [1,*], Dawid Simiński [2], Mikołaj Kruszewski [3], Anna Niedzicka [3], Kamila Gąsiorek [3], Aliya Batyrbekovna Zabieva [4], Jarosław Mamala [5] and Ewa Dębicka [3]

[1] Faculty of Transport and Aviation Engineering, Silesian University of Technology, 40-019 Katowice, Poland
[2] Doctoral School, Silesian University of Technology, 44-100 Gliwice, Poland; dawid.siminski@polsl.pl
[3] Motor Transport Institute, 03-301 Warsaw, Poland; mikolaj.kruszewski@its.waw.pl (M.K.);
  anna.niedzicka@its.waw.pl (A.N.); kamila.gasiorek@its.waw.pl (K.G.); ewa.debicka@its.waw.pl (E.D.)
[4] Faculty of Transport and Energy, Eurasian National University, Astana 010000, Kazakhstan;
  aliya.zhakupovazabieva@gmail.com
[5] Department of Vehicles, Faculty of Mechanical Engineering, Opole University of Technology,
  5-271 Opole, Poland; j.mamala@po.edu.pl
* Correspondence: rafal.burdzik@polsl.pl; Tel.: +48-32-6034116

**Abstract:** The paper presents a proposed methodology for designing and planning research on driver behavior at pedestrian crossings using whole-vehicle simulators. It was assumed that dedicated research should be conducted in specific problem contexts. The problems identified were the identification of hazards and the risk of accidents involving vulnerable road users. The purpose of this identification is to determine the determinants of safety at pedestrian crossings, which should constitute guidance when designing new solutions for safety support systems at pedestrian crossings. A number of hazard factors were identified; divided into categories, including types of crossings, location, and surroundings; behavior of vulnerable road users; and attention (focus) distractors, both inside and outside the vehicle. A method for defining and selecting research scenarios and selecting a group of research participants was proposed. Additionally, it was proposed to conduct repeatable test scenarios for different driving speeds and different weather conditions. With respect to the publications on this topic, this work focuses on the process of designing and planning dedicated simulation studies, which may provide a source of guidance and good practices for other researchers. This is an example of how interdisciplinary research involving human factors, traffic organization, and ITS systems can be planned and implemented.

**Keywords:** pedestrians; drivers; driving simulators; road safety; pedestrian crossings

## 1. Introduction

The issue of safety in road transport is crucial and multifaceted. Many factors can be distinguished that influence this safety to a greater or lesser extent. One approach to analyze transport safety is to identify the hazards and assess the risk of an accident or road traffic collision [1,2]. The mechanisms and habits that define the set of behaviors of traffic participants, including drivers and pedestrians, constitute one of the more important groups of risk factors [3]. The difficulty of analyzing and assessing them stems from the specialized nature of behavioral research into human behavioral mechanisms [4]. Current research in this area becomes even more relevant in the context of the development of autonomous transport and the achievement of fully autonomous motor vehicles [5,6]. This study [7] outlines the key issues involved in the development of hybrid and electric vehicles, issues unknown to conventional vehicles with internal combustion engines. The growing number of electric and autonomous cars [8], which operate quietly and with minimal vibrations, requires a re-evaluation of safety factors. Noise and vibration are natural indicators of potential hazards [9,10].

The issue of driver behavior is very important, especially with regard to the consequences that may arise for vulnerable road users, i.e., the risk of traffic accidents. Despite many driver education campaigns [11], it is assumed that driver behavior is a contributing factor to the majority of road accidents worldwide, so it is extremely important to have as much knowledge as possible about the various factors and their impact on the possibility of a road accident occurrence. This topic is so important that, according to the World Health Organization, there are approximately 1.3 million fatal accidents and between 20 and 50 million accidents [12] involving drivers and vulnerable road users each year. According to the accepted standards, there are different methods of data collection used to analyze individual driver profiles. One of the methods that can be used is the demographic profile binning method [13]. Other methods are those involving the use of different types of tests, questionnaires, psychological interviews, and other tools related to learning about drivers' behavior according to what they themselves declare in terms of the risks they are willing to take when driving a motor vehicle in relation to overall road transport safety [14]. In addition, it is possible to collect data by reading recorded information from devices on the vehicle. This type of study offers the possibility of tracking driver behavior not only at the time of the measurement itself, under controlled conditions, but offers the possibility of a continuous, uninterrupted cycle of data collection [15]. In addition, observational studies are also performed and individual situations archived via video cameras are analyzed. Responses to road situations may be examined using computer simulators [16]. Methods using simulation devices—simulators—aim to replicate, as closely as possible, the real-life conditions in which individual situations can occur. However, there is a safety buffer in the tests that are carried out in a safe manner, excluding the risk factors associated with the fact that health or life will be endangered in the event of a non-compliant response, as may occur in normal traffic. It also involves the responsibility of those who carry out such tests. It would be very difficult under real conditions to study the reaction of drivers during situations such as skidding in normal traffic conditons, which could have fatal consequences, such as a road accident. On the other hand, however, conducting tests under controlled conditions to the full extent may be associated with a lack of authenticity of the acquired data [17]. This is why the planning and design stage of simulation studies is so important and directly affects the quality and usability of the data obtained and the resulting validity of the conclusions drawn.

The aspect of authenticity of the study was also very important during the research planned at the Motor Transport Institute in Warsaw concerning the behavior of drivers approaching pedestrian crossings, in different variants of pedestrian crossing markings, such as classic vertical signs, luminous vertical signs, and luminous horizontal signs, occurring in different configurations. The simulator facilities used at the Motor Transport Institute in Warsaw are a highly advanced technology for carrying out extensive analyses of the behavior of drivers of not only passenger vehicles, but also heavy goods vehicles in near-real conditions, reproducing in a simulated environment the same reactions as driving a motor vehicle in normal road conditions.

In addition to the passenger car simulators described in this study, we can distinguish, for example, simulators for military applications, where the training of soldiers or service personnel takes place under controlled conditions in which there is no likelihood of a catastrophe associated with the improper handling of equipment or vehicles. The same applies to flight simulators, where pilots undergo an extensive training process. The procedures that take place during such training are the same as those that take place in real-world conditions. However, in addition to simulators that are so advanced, there are other simulator solutions that provide variety and a very high degree of innovation during training [18]. At the TRAKO 2023 trade fair organized in Gdańsk, the simulators used during the training process for employees of the Polish Railway Lines were presented, an example being a simulator for operating a platform for disabled people on electric multiple units [19]. This makes it possible to conduct a range of training courses in a minimally invasive way, with the aim of assimilating appropriate operating procedures.

This study addresses a very complex issue of road safety, in particular, relating to the behavior of both vulnerable road users and drivers of motor vehicles. In the case of the tests carried out directly in traffic, a driver performing particular tasks aimed at testing his or her alertness and ability to assess the situation under given conditions would be exposed, as well as endangering other road users, which is of course unacceptable. This is why the development of highly sophisticated driving simulators was such a breakthrough achievement. They offer the capability to replicate real-world conditions in a simulated environment, ensuring safety in the event of a collision or accident.

Currently, there is a dynamic development of innovative pedestrian crossings with specific parameters [20]. However, they can also contribute to traffic congestion in certain areas. These crossings not only enhance road safety, but also improve traffic flow, thereby reducing congestion [21]. Pedestrian crossings are crucial for ensuring the safety of road users and are also an integral part of road infrastructure. Autonomous or semi-autonomous cars pose a new group of hazards in transport, particularly in terms of pedestrian safety at crossings [22]. Despite obstacle recognition and tracking devices becoming increasingly sophisticated and aided by AI (artificial intelligence), there is still little public confidence in these vehicles.

This study presents the complete process involved in conducting simulator studies, from general examples defining the framework related to the simulator study approach to ready-to-use guidelines with study scenarios, based on complex analyses using behavioral studies of both drivers and vulnerable road users, together with the determination of safety determinants of pedestrian crossings without traffic lights. Due to the specific nature of simulator testing and the frequent complaints during long-duration testing, a reduction in testing time while maintaining a representative number of cases and traffic scenarios to be simulated in order to identify driver behavior and perceptions that may influence the risk of a pedestrian crossing accident was adopted as a key element in the planning and design of the study. Thus, a comprehensive method and algorithm for planning and designing simulator studies should be considered as the main achievement presented in this study.

## 2. Principles of Simulator Testing

When starting the research process, the most important element is the definition of objectives. This is a key aspect that will determine the further relevant steps of the process. During preparation, it is necessary to precisely identify the problem that will be verified during the study in question or, in the absence of a positive result, will not be clarified. At this stage, it is important to analyze the current state of knowledge in a given issue and make a synthetic comparison of what has been achieved to date. The next step in carrying out simulator-based research is to choose the right equipment to meet the research objectives, namely the choice of appropriate driving simulators and the configuration of the measurement and control system. It is important that, in the case of work relating to perception studies in normal traffic, the result should be as authentic and realistic as possible. Therefore, when analyzing studies by other researchers, it is possible to make use of the knowledge that has emerged concerning studies directly related to recorded driving events using simulator-built equipment, but also questionnaire studies that provide insights into the research process and its authenticity [23].

Types of Driving Simulators for Passenger Cars

Driving simulators are primarily tools that provide the opportunity to learn about individual behavior in a safe simulated environment, but they also serve to educate drivers in a broad sense [24]. Depending on the complexity and relevance of the research carried out on the use of driving simulators for passenger cars, we can distinguish several types of driving simulators, depending on their functionality and technological advancement. The basic division of driving simulators can begin with so-called augmented reality simulators, very often used for various types of experiments or for training drivers, pilots, and operators [25]. The next group is made up of desktop simulators. These are basic devices that provide the ability to recreate the simulated environment within a very limited range

of operation. Compact simulators are the next type, providing greater capabilities than desktop simulators, as they implement, among other things, additional elements, such as seats similar to the ones installed in real vehicles. Another group is whole-vehicle driving simulators, which are mainly used in universities and research centers, as they reflect the real environment in simulated conditions, i.e., the ability to drive a vehicle in a manner similar to that of a motor vehicle in operational conditions. The last group relates to solutions at the disposal of research and development institutes conducting very advanced research into driver behavior while driving mechanical vehicles, equipped not only with a real-size cab and equipment, but also with several degrees of freedom of movement; such simulators are also called dynamic. The sensation of driving this type of driving simulator is close to that of driving an actual motor vehicle. Depending on the sophistication, differences for the individual simulators may include the angle of the field of view, image resolution in pixels, image brightness, contrast, and the ability to combine images from several projectors at once. The latency of the scene display and the delay of the motion system after input by the driver are also of great importance. Further differences relate to the animation of the motion scene and the quality of the displayed content. The question of the realism of the measurements is also very important by adjusting, for example, the sound connected to the vehicle actuators accordingly. All these issues vary depending on the apparatus used. While the test results from simulators with a minimum of six degrees of freedom are able to reproduce the driving conditions of a real car in a very realistic way, the use of a static desktop simulator does not guarantee such effects [25].

## 3. Simulator Sickness

The phenomenon known as simulator sickness is very often unavoidable when conducting research work on any type of equipment that mirrors real-world conditions in a controlled virtual environment [26]. The most common symptoms of this phenomenon include fatigue, drowsiness, dry mouth, excessive saliva production, nausea, disorientation problems, dizziness, and stomach complaints [27]. Symptoms of simulator sickness are assumed to occur when information from all the senses that aid in spatial orientation and movement perception conflict with what the person's previous experience was. It is assumed that, during the execution of individual movements in a simulated environment, the pattern of information that was previously performed in the natural environment is quite different compared to that performed on the simulator [28]. The incompatibility occurring between the sensory information received from the simulator and that which was anticipated results in the occurrence of a phenomenon known as simulator sickness [29]. The phenomenon of simulator sickness can have a profound impact on the testing process itself and the results. When a person is susceptible to the influence of simulator sickness, the testing process can be disrupted. Inference from the results disturbed by this phenomenon can lead to erroneous conclusions. In training, this can lead to negative results [30]. Research using the SSQ (Simulator Sickness Questionnaire) [31] addressed simulator sickness symptom severity depending on the visual conditions present. To be able to compare subjective feelings, the Simulator Sickness Questionnaire can be used. An exemplar of a developed questionnaire for the assessment of perceptibility is presented in Figure 1. The results of the study [31] show that the use of displays directly placed over the eyes as VR goggles results in as much as a 70 percent increase in simulator sickness symptoms compared to the use of static conditions in the simulator study, where the screens were significantly distant from the subject [31].

Another study indicating the severity of simulator sickness symptoms was described in [32]. They investigated the effect of the type of simulator and its purpose on the severity of simulator sickness symptoms. The studies conducted focused on the use of aircraft, helicopter, and passenger car simulators. Interestingly, the greatest negative effects associated with the occurrence of simulator sickness occurred in the case of the passenger car simulator; here, serious disorientation problems were observed in the research participants. In the case of the helicopter and aircraft simulators, the highest levels of discomfort

were related to oculomotor disorders. A very interesting phenomenon was observed by researchers focusing on the movement capabilities of the simulator [32] investigating the possibility of a difference in feelings of well-being depending on the type of platform on which the simulator was installed. The severity of symptoms of simulator sickness was greater in subjects on a static platform compared with a platform equipped with six degrees of freedom [33]. In view of the above-mentioned examples of research related to the use of different types of simulators, and in particular the fact that, according to a study, when comparing different types of simulators, it was found that the incidence of simulator sickness was highest when a passenger car driving simulator was used. This information is crucial for the planning of this type of research, and together with studies [32,33], indicates the direction that should be taken into account when choosing a suitable testing device. Due to the fact that symptoms of simulator sickness increased by up to 70% among participants in studies using VR [33], it was decided not to use VR (virtual reality) goggles in the planned study in this case, so as not to expose the study participants to a high percentage of the risk of simulator sickness; given that the individual route is planned to be driven several times, this could be an element that would eliminate individual participants from taking part in the study. According to the study [33], it was decided to use a simulator device equipped with six degrees of freedom due to the lower probability of simulator sickness symptoms, as well as the greater realism of the scenario trials themselves [33].

**SIMULATOR SICKNESS QUESTIONNAIRE**

Circle how much each symptom below is affecting you now.

0 = "not at all"   1 = "mild"   2 = "moderate"   3 = "severe"

| | | | | | |
|---|---|---|---|---|---|
| 1. | General discomfort | 0 | 1 | 2 | 3 |
| 2. | Fatigue | 0 | 1 | 2 | 3 |
| 3. | Headache | 0 | 1 | 2 | 3 |
| 4. | Eyestrain | 0 | 1 | 2 | 3 |
| 5. | Difficulty focusing | 0 | 1 | 2 | 3 |
| 6. | Increased salivation | 0 | 1 | 2 | 3 |
| 7. | Sweating | 0 | 1 | 2 | 3 |
| 8. | Nausea | 0 | 1 | 2 | 3 |
| 9. | Difficulty concentrating | 0 | 1 | 2 | 3 |
| 10. | Fullness of head | 0 | 1 | 2 | 3 |
| 11. | Blurred vision | 0 | 1 | 2 | 3 |
| 12. | Dizziness (eyes open) | 0 | 1 | 2 | 3 |
| 13. | Dizziness (eyes closed) | 0 | 1 | 2 | 3 |
| 14. | Vertigo | 0 | 1 | 2 | 3 |
| 15. | Stomach awareness | 0 | 1 | 2 | 3 |
| 16. | Burping | 0 | 1 | 2 | 3 |

**Figure 1.** An example of SSQ (Simulator Sickness Questionnaire) content.

## 4. Planning and Design of Studies of Driver Behavior at Pedestrian Crossings Using Simulators

A key aspect when planning and designing studies using driving simulators is their context. A study is planned differently in the context of, for example, the impact of driving speed, to address the impact of fatigue, and in the case of monotony. One of the most dangerous effects of driving in a monotonous manner is the phenomenon of microsleep [34]. The context of the research may also relate to specific group characteristics of the participants, e.g., age, gender, and health issues. One of the context groups is behavior for specific traffic scenes, e.g., driving in congested conditions in the city, at level crossings, and various types of intersections. All these aspects influence the need to select the type of driving simulator, define the test scenarios, determine the preferences of the users (test subjects), select the measurement parameters, and organize the research process. The context of this research concerned the behavior of drivers in the vicinity of pedestrian crossings and involved a general group of users.

### 4.1. Defining the Research Context and Assumptions

The research problem and context determine the planning, design, and implementation processes. Very often, a precise and accurate formulation of the research context makes it possible to significantly reduce the scenarios and conditions, and thus the scope of the research, and allows a significant shortening of the execution time while maintaining the quality of the measurement results obtained. Two key research problems were identified as factors affecting drivers' distractions (distractors inside the vehicle, such as operating touch panels) and systems for signaling drivers when approaching a pedestrian crossing, including interactive innovative signs and pictograms.

In this study, the research problem is the behavior of drivers in the vicinity of different pedestrian crossings with different factors influencing drivers' perception and focus. The research problem defined in this way made it possible to define a research context, which involved following drivers' behavior and perceptions when approaching pedestrian crossings under different driving conditions, defined by vehicle speed, weather conditions, and the type of surrounding infrastructure. In addition, the context included various assumed specific behaviors of vulnerable road users (turning, accelerating step, etc.) and different types of pedestrian crossing infrastructures and equipment with warning systems. The influence of internal and external attentional distractors on driver behavior, such as talking on the phone or operating a screen in the vehicle or on a tablet (phone), as well as roadside advertising or unusual objects outside the vehicle, was also identified as an important component of the research context. The general types of such attention distractors, based on [31], are shown in Figure 2.

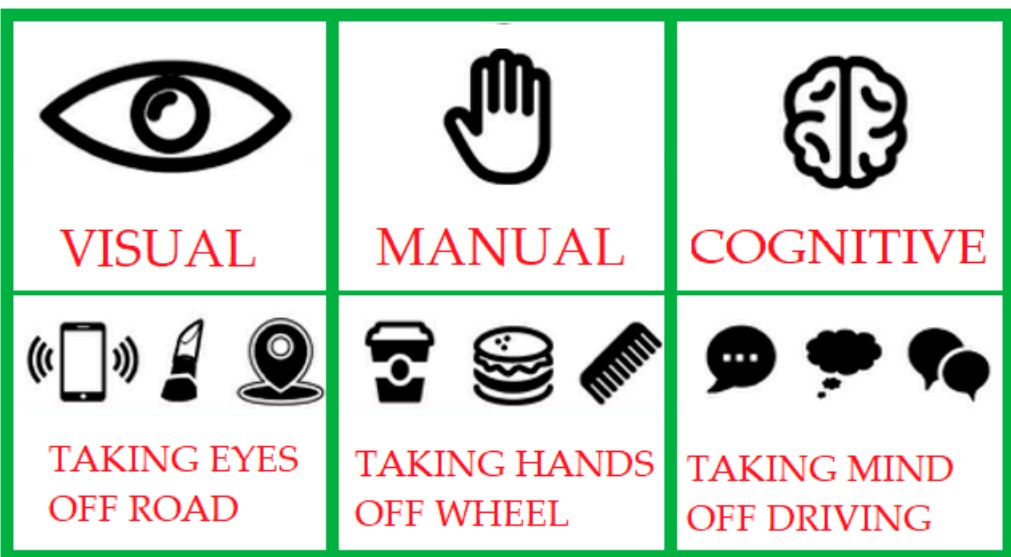

**Figure 2.** Visual, manual, and cognitive distractors affecting the driver.

As a result, the research assumptions related with detailed research problems were defined as:

- Analysis of the impact of the type of pedestrian crossing;
- Analysis of the impact of the method (system) of warning;
- Analysis of the impact of surrounding infrastructure conditions;
- Analysis of the impacts of driving speed and weather conditions (including time of day);
- Analysis of the impact of the behavior of vulnerable traffic participants (pedestrian and cyclist);
- Analysis of the impacts of internal and external distractors on attention.

Additional assumptions limiting the scope of the study were:

- Elimination of the effect of driver fatigue during the study (short testing time);

- Elimination of the effect of additional medical conditions on driving ability;
- Eliminating the impact of driving experience;
- Eliminating the impact of road infrastructure lighting and driver glare;
- Eliminating the impact of additional vehicle safety systems (including "assistances" and obstacle detection sensors).

*4.2. Selection of the Type of the Driving Simulator*

The selection of the type of driving simulator included several aspects. One of them was the main purpose for which the research is to be carried out, as it is to study the behavior of drivers approaching pedestrian crossings, and the results can then be used to create solutions to support the safety of vulnerable road users. The main requirement was first and foremost the realism of the research itself, to replicate real conditions in a simulated environment as much as possible. Another aspect concerned its complementarity with regard to the functionality of the device. When planning a behavioral study, research parameters are determined that will be crucial to the simulation study. The research parameters in the case of driving simulator research may be: driving speed, any kind of reaction time relating to the devices in the driving simulator, eye tracking, and movement of the research participant in the vehicle. The final selection of the phenomena to be studied and the resulting measurement parameters are always determined by the specific context of dedicated research. When identifying hazards at pedestrian crossings, it is crucial to consider the driver's perception of the traffic scene, concentration, and correctness of reaction. The factors influencing these observations should include driving speed, behavior of vulnerable road users, attention distractors, and weather conditions. When analyzing these factors, it is necessary to determine the measurement signals recorded during tests. Following an analysis of the available driving simulators and consideration of the context and research problems, it was determined that the high-end simulator AutoSim AS 1200-6 (AutoSim AS, Tromsø, Norway) was an appropriate choice. The high-end simulator AutoSim AS 1200-6 is built from a full-size cabin of an Opel Astra IV passenger car and equipped with a system of four projectors displaying images on a screen covering a 200° field of vision, as well as a system of three screens acting as car mirrors (Figure 3). The simulator sits on a six-degree-of-freedom moving platform, allowing the simulation of angular and linear movements of the cabin during actual driving. The simulator's software (v.2016.10) allows full control of weather conditions, lighting conditions, and the behavior of other road users. An additional advantage of the solution is the minimization of the risk of simulator sickness by not using directly worn VR goggles. The simulator has a number of relevant cameras to record activity in the cab, and it is also crucial that the simulator has six degrees of freedom, which also has a great impact on the ability to minimize the occurrence of symptoms of simulator sickness among study participants, which is crucial for reliable results after the completion of the study, as extensively described in the previous section of the article. The last aspect concerns the availability of such an advanced solution. Simulators of the class described in this study are available for use only at specialized research institutes.

The selection of the type of driving simulator is a key stage of research planning and must result from the specific research problem and scope of experiments. In the case of research focusing on, for example, reaction time, driving bucks are often used as cheaper solutions than professional full-vehicle cabin simulators. These types of simulators are often additionally equipped with actual adjustable seat bucks with a head-mounted display (HMD) as a visualization system [35]. Desktop simulators or even VR are being used more often because they are definitely the cheapest methods [36]. However, significant differences in this case include even the observed motion scene and surface area. However, the most important difference in these solutions is the lack of perception of dynamic phenomena (acceleration, braking, tilting, and even collisions) [37]. It is true that driving bucks equipped with moving platforms are available, but the sensation of these phenomena is different compared to a closed car cabin. However, it should be remembered that the

dynamic technological development in the simulator and VR sector continuously improves the "reality" of these solutions, and due to the cost disproportion in relation to dynamic full-vehicle simulators, more and more studies will use such solutions.

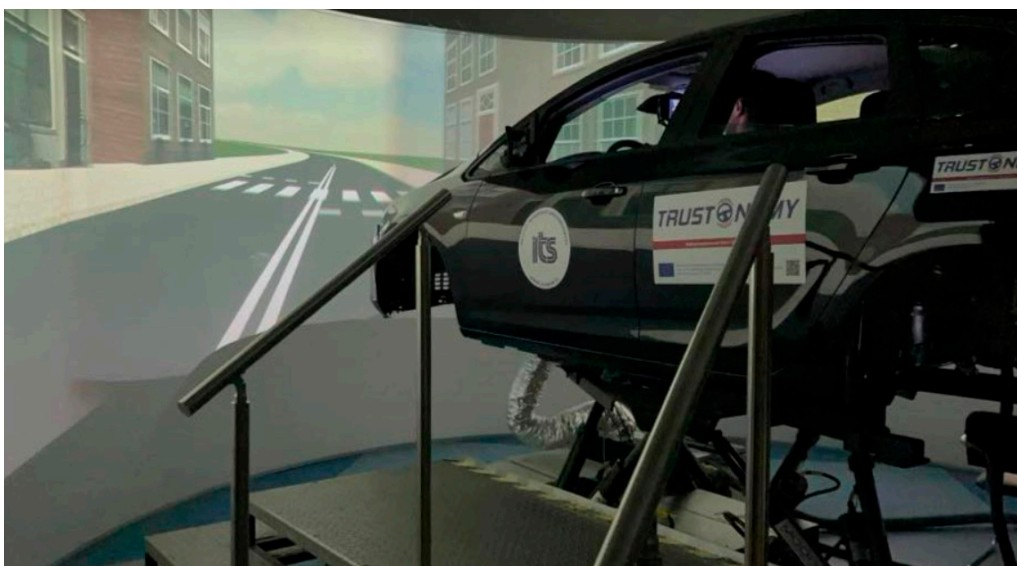

**Figure 3.** AS 1200-6 passenger car simulator (source: Motor Transport Institute Warsaw; own picture).

*4.3. Research Planning*

Once the problem and scope of the research have been defined and the research methods and equipment, including the type of simulator, have been selected, the planning stage of the research should begin. If the research involves the participation of people, it is necessary to identify specific preferences, such as age, gender, health condition, work experience, and many others. In this case, it is imperative to form an opinion on the possibilities and limitations of the research with human participants before the relevant ethics committees. Once approval has been obtained, consideration should be given to the possibility of conducting additional pre-tests that allow additional information on the personality traits of the research participants to be obtained. For the case of simulation research, all participants should sign a consent form and a statement that show they are familiar with the possible consequences of taking part in the research using driving simulators [38]. For this purpose, the participants can complete the SSQ (SSQ1) [31] and later begin an adaptive driving session.

Planning simulator research involves a number of activities, starting from defining the assumptions and context of the study, through the selection of measurement apparatus, the determination of preferences and recruitment of the research group selection of research scenario configuration of simulator software, testing and verification, and final scheduling of the core study. A general flowchart of the planning process for dedicated simulator studies is shown in Figure 4.

All the activities and operations identified on the basis of the work breakdown structure (WBS) and the estimated time for their implementation are presented in Table 1. Some of these operations can be carried out simultaneously. Then they are assigned a start-to-start or finish-to-finish relationship (e.g., operations 6 to 10). However, this always requires a detailed analysis to ensure that even small changes in these activities do not affect the execution of parallel tasks.

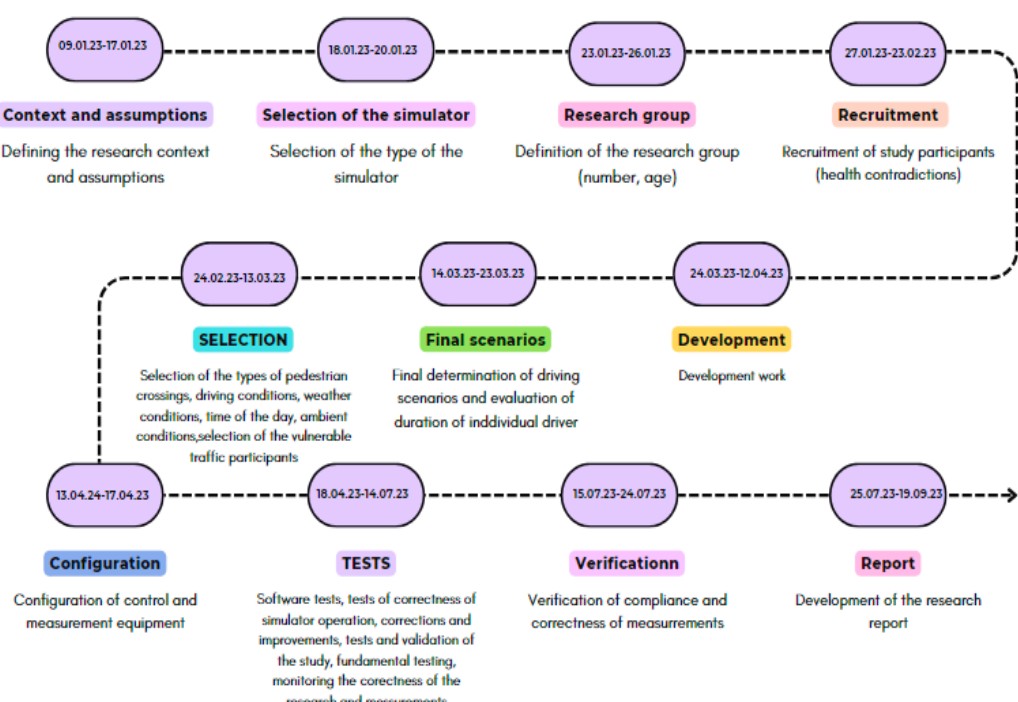

**Figure 4.** Flowchart of research planning process (CANVA software (canva.com, accessed on 12 November 2023)).

**Table 1.** List of operations and time duration of dedicated simulation research planning process.

| Operation | Time Duration |
|---|---|
| 1—Defining the research context and assumptions | 1 week |
| 2—Selection of simulator | 3 days |
| 3—Definition of the research group (number and age, etc.) | 1 day |
| 4—Recruitment of study participants (verification of health contraindications) | 3 weeks |
| 5—Selection of context (e.g., types of pedestrian crossings) | 1 week |
| 6—Selection of driving conditions (e.g., speed and location) | 1 day |
| 7—Selection of weather conditions | 1 day |
| 8—Selection of times of day and night of research | 1 day |
| 9—Selection of ambient conditions (built environment) | 1 day |
| 10—Selection of vulnerable traffic participants | 1 day |
| 11—Final determination of driving scenarios—3 days | 3 days |
| 12—Estimation of the time of individual driving stages and the total study | 3 days |
| 13—Development work (design of the study according to the guidelines) | 2 weeks |
| 14—Configuration of control and measurement equipment (determination of recorded data) | 3 days |
| 15—Software tests | 3 days |
| 16—Tests of correctness of simulator operation | 5–7 days |
| 17—Corrections and improvements—1 week | 1 week |
| 18—Tests and validation of the study (evaluation of compliance with the assumptions, purpose, and scope of the basis) | 5 days |
| 19—Core research (according to the accepted scope) | 2–6 weeks |
| 20—Monitoring the correctness of the research and measurement process (in parallel with 19) | 2–6 weeks |
| 21—Verification of compliance and correctness of measurement results | 1–2 weeks |
| 22—Development of the research report | 4–6 weeks |

The research planning process, carried out in this way, makes it possible to identify all the necessary operations and estimate their duration time. Ultimately, this makes it possible

to precisely determine the time required to carry out the entire study and to develop a schedule using, for example, a Gantt chart (Figure 5).

**Figure 5.** Example of schedule for planning and execution of dedicated simulator studies in the form of a Gantt chart (Gantt software v3.2).

## 4.4. Defining the Research Scenario

Defining the test scenario is a fundamental step when preparing to perform any test. This stage is labor-intensive due to the multitude of different traffic situations that can occur in real conditions when moving a vehicle. The purpose of selecting individual test scenarios is to select as many configurations as possible of a particular travel route as needed. It also refers to establishing the initial and variable conditions to be controlled. Driving simulators provide an unprecedented opportunity for the repeatability of implemented conditions, which provides very tangible benefits when performing individual measurements for different groups of people depending on their age or gender. Naturalistic, real-world conditions do not offer the possibility of such a selection of ideal external conditions, because there is a high risk that, for each ride, these conditions will be slightly different. In the case of simulators, the conditions are analogous and repeatable for each participant taking part in the study. The selection of the research scenario also reflects the most relevant needs during the conduct of the study, as the number of possible configurations is arbitrary. After the initial preparation of the extensive possibilities of scenarios, the most relevant are selected due to the nature of the work, but also these limitations relate to the amount of time that the individual participant is able to focus and actively participate in the study. A major threat when conducting this type of research is the phenomenon associated with simulator sickness, a phenomenon that is described in the previous section.

For this purpose, an approach of developing reciprocity tables is proposed, which are created on the basis of a square matrix containing all the required specifications of the research scenarios. As a result of the analysis, it is possible to eliminate repeated combinations and then group the remaining scenarios to optimize the final number of scenarios based on the target function defined as the allowable (maximum) execution time of a single study.

The work on defining the research scenarios began by performing a series of studies related to behavioral factors and on the basis of safety determinants. The behavioral research involved analyzing the behavior of a group of 900 drivers approaching pedestrian crossings without traffic lights. The authors conducted these studies in the form of continuous and blind observations using a developed observation protocol and additionally video recording for the post-processing analysis. The observed traffic participants were unaware that they were being observed, thus exhibiting natural behavior. Using the appropriate research apparatus, microwave radar, it was determined how the group of drivers surveyed behave when approaching the pedestrian crossings. The measurement was carried out at a distance of 30 m in front of the pedestrian crossing up to the crossing itself. The conditions that were taken into account during the study included weather conditions, time of day, and

season, so that the results could be compared for different naturally occurring conditions throughout the year. The observation sheet on the current behavior of drivers approaching pedestrian crossings concerned the recording of information on whether the driver slowed down, accelerated, or maintained a constant speed while crossing the pedestrian crossing. This was followed by information regarding the driver's use of any kind of distraction or performing undesirable activities while driving. These behaviors included talking on the phone or using the phone for activities other than talking, which raises some theories regarding cell phone use [34,39]. This was followed by using a screen in the vehicle, talking to another passenger, looking around while approaching a pedestrian crossing, eating or drinking, smoking tobacco products, or performing other unspecified activities in the vehicle that may increase the risk of a traffic accident. The process of classification into gender and age groups of drivers was determined in accordance with official statistics published by the Police Headquarters. Subsequently, statistical tests were performed on the collected data on the occurrence of accidents at pedestrian crossings. These analyses involved the determination of 100 accident localizations, the so-called black spots. The analysis consisted of selecting accident sites from various places in the country, divided into provinces with the highest and lowest numbers of accidents at pedestrian crossings, then the Silesian province was also included in this group, as well as individual sites from across the country with the highest number of accidents at pedestrian crossings nationwide. After determining the most sensitive places from the point of view of traffic safety in relation to vulnerable traffic participants, an extensive analysis was carried out of the places where the accidents took place, with a description of the events in the traffic accident database as a confirmation of authenticity. The analysis dealt directly with the places where the incidents took place. It described possible causes related to the traffic situation. Since the information on the direct cause of traffic accidents is not public, as it is based on witness statements or the perpetrators directly, it is not possible to determine the direct cause of a traffic accident, so by performing an analysis of the sites, it is possible to identify recurring elements of the pedestrian crossing infrastructure that can affect the occurrence of an incident. This way, the determinants related to the infrastructure safety of pedestrian crossings were identified. Then, with reference to the secrecy of the direct causes of accidents, an extensive survey was conducted in relation to the direct perpetrators of traffic accidents. The survey included a series of questions related to the causes of traffic incidents. The survey, performed anonymously, approximates the elements that can affect the occurrence of a traffic accident; hence, the results were used to extract a group of scenarios associated with possible dangerous situations at pedestrian crossings.

### 4.5. Configuration of the Set of Recorded Data

Configuration of the set of recorded data from the measurement apparatus is a process that should be determined at the research planning stage. Configuration of the simulator according to the assumptions of the research process is crucial, because when formulating the assumptions, it may turn out that the main research problems cannot be studied due to the limitations of the apparatus that will be used for the research. Equipment limitations determine whether a research unit will be considered for selection in the market research process. In the context of driving simulator research in Poland, there is a very narrow group of research units capable of conducting advanced research programs, which limits the freedom to conduct this type of research on a large scale, thus making it very specialized research, also due to the limited number of specialists with expertise in this area.

At this stage, it is very important to define and group driver reactions and behaviors in detail, and to determine the parameters that characterize them, e.g., reactions and perceptions. The control and measurement apparatus consist of many sensors that record specific signals [40]. It is difficult to design and manufacture a system that will directly measure reaction or perception. Currently, such measurements can, admittedly, be implemented almost directly using machine learning methods. The use of artificial intelligence algorithms that can classify behavior on the basis of recorded signals is becoming more and

more common, but still depends on the decisions of the neural network, which we do not always have an influence on as operators of such a system. Another approach is to try to define groups of behaviors or perceptions based on available measurement signals, such as the timing of brake application or saccadic movements or gaze fixation. In this way, by configuring the measurement system accordingly, we are able to record indirect data, such as behaviors and perceptions, using other directly recorded signals. Often, this stage of research is carried out in "post-processing", at the results analysis stage.

In the majority of studies on driver behavior, reaction time is considered to be the primary indicator. In order to gain a more comprehensive understanding of reaction time, it is necessary to consider the various components that contribute to it. These include perception, visibility, and intuition. Study [15] presents an innovative method of research on the perception and selection of the message of the variable message sign. The authors employed multi-channel sets of recorded signals, eliminating their mutual dependencies. The research employed a range of signals, including those derived from the CAN bus and independent driving dynamics signals, such as vibration acceleration, eye tracking, and a volitional acoustic channel. In the case of reaction time testing, the set of dependent variables is extensive and crucial, starting from vehicle speed through attention-related variables to scenario-specific factors. The assumptions and purpose of the research, as outlined in this study, are to identify risk factors when passing pedestrian crossings of various types and in various conditions, with the participation of different vulnerable road users. Accordingly, the dataset should include, among other variables, the type of distractor present in the vehicle, the type of crossing, the location and surroundings of the crossing (external distractors), the type of crossing marking (including innovative solutions), various types of pedestrian behavior, as well as driving speed and weather conditions. In order to assess driver behavior, it is proposed that the field of vision (eye tracking), reaction times as a function of distance from pedestrians, as well as driving parameters and collision situations can be recorded. These measures can be grouped according to specific defined pedestrian crossing types and their locations in order to observe and compare safety at different pedestrian crossings.

*4.6. Research Ethics*

Ethical and safety risks associated with conducting simulator tests are key to ensuring, among other things, the anonymity of those taking part in simulator research. Individuals taking part in this type of research must be aware that their data will not be disseminated in an uncontrolled manner, and hence, in order to maintain ethics, it is not advisable to ask broadly sensitive questions of individual participants when conducting surveys. It is also crucial to ensure safety. Subjects must not be exposed to the dangers of possible consequences associated with malfunctioning mechanical devices. Thus, the driving simulator cab was equipped with a number of actuators responsible for changing the position of the vehicle during driving to properly replicate the sensation of driving, like in a conventional motor vehicle. Of course, any mechanical device can be a potential danger to health or life, so participants must be informed of the potential dangers before the study. When recruiting participants for the study, an extensive interview relating to the individual's possible medical contraindications was carried out, with the consent of the subjects themselves. This is solely related to the risks that may take place during the research process associated with the use of the driving simulator. The recruitment of participants was carried out taking into account all these factors, as well as the research assumptions of the study itself, age, gender, and other parameters, such as visual defects.

All research involving humans must be approved by the appropriate ethics committee. When planning simulator tests, it is essential to assess human exposure during the tests, as well as any medical and other contraindications resulting from personality traits and even worldview. Potential research participants must be informed during recruitment about all the risks and potential consequences of the research, and their full consent must be obtained.

### 4.7. Participant Training

Before starting the process of taking measurements on the driving simulator, detailed training should be performed related to the transfer of all the necessary information for participants about the operation of the simulator. It is necessary to generally present the scope of the test itself, what it will consist of, but without details that may affect the decisions made during the test itself. The training is more about making the person taking part in the study feel comfortable while driving in the simulator, and making sure that they understand what tasks they have to perform. The aspects of understanding the idea itself and the correct approach to the tasks performed are key, because it often happens that participants have not properly assimilated the knowledge during training, but when asked, they claim they understand, even though their knowledge of the procedure itself is incomplete. It is worth performing a preliminary study, which will indicate the aim of the study without pointing out the aspects relevant to the research objectives. This will make full use of the research participants' time and eliminate potential inappropriate measurements that will have to be discarded during the analysis, and considering the financial aspect, it is recommended to take full advantage of the research opportunities. After the several-step preparatory process, once the research process has begun, it is important to implement the research according to a developed scenario in order to maintain consistency. Based on the data recorded by the equipment installed in and outside the vehicle, it is also worth conducting an independent external analysis, record observations, and unusual behavior in particular situations, recorded during subsequent drives made by the test participants. These will be a kind of supplement to the knowledge gained through the measurement equipment on the vehicle. The process of conducting the actual tests is followed by the process of analyzing the data, along with the interpretation of the obtained results. These involve carrying out a statistical analysis in accordance with the initial assumptions before the start of the research; the interpretation and inference of the obtained results will take place in the context of the set research objectives.

Once the research process is completed, it is important to prepare complete, comprehensive analyses that relate directly to all the steps preceding the research process. The publication of the obtained results provides an opportunity to expand the knowledge of the research team, but also of those conducting scientific activities in the given subject area.

### 4.8. Research Context Implementation—Attentional Distractors

One of the objectives of the study was to analyze the impacts of internal and external attentional distractors, which strongly affect the process of perception of essential information, i.e., the traffic scene and the level of concentration of drivers. We embarked on a simulator study, one of the goals of which was to obtain information on the perceptibility and behavioral evaluation of drivers when they were presented with implemented pseudo-random distractors, such as N-back and SuRT tasks (Figures 6 and 7). These tasks affect the ability to focus on the activity and can significantly affect the results of the study. The N-back task involves verbally repeating a sequence of numbers in sequence read by a recorded voice. In the 0-back condition, the participant had to repeat the last number in the sequence; in the 1-back condition, the participant repeated the penultimate number. The SuRT test is a test depending on the level of difficulty involving the participant in a significant way; this task involves pointing at a tablet mounted on the simulator, on the right side of the driver, at the level of the function panel in the vehicle. The participant is asked to point to the circle with the largest diameter when a group of circles appears on the tablet screen. The participant is not immediately informed whether the circle he or she pointed to is the correct one; the results are analyzed only after the test is completed. Depending on the difference between the diameters, it may be significantly more difficult for the participant to find the largest circle and makes them more involved in the test, which is associated with the risks while driving.

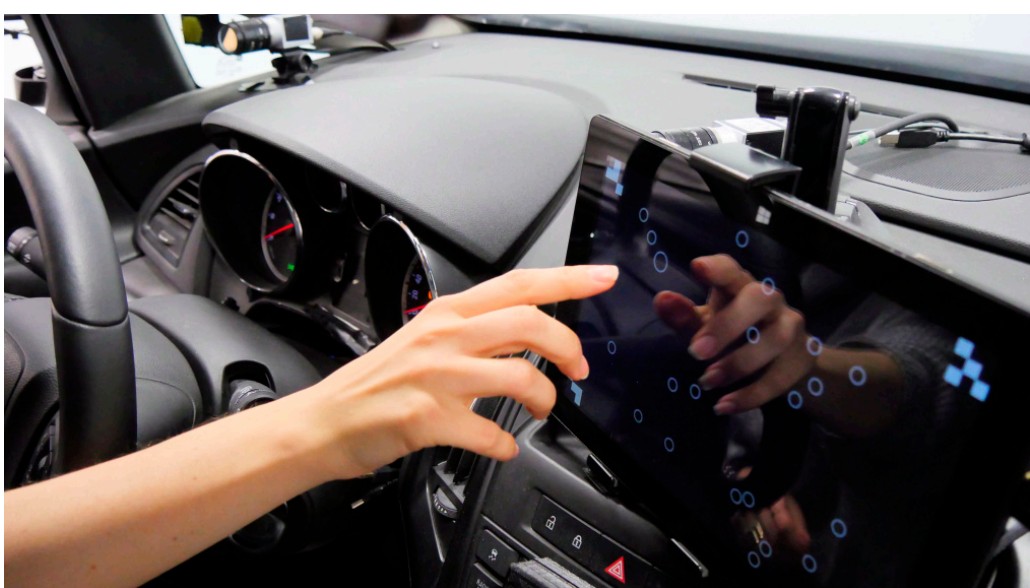

**Figure 6.** Overview photo for the SuRT task (source: Motor Transport Institute Warsaw; own picture).

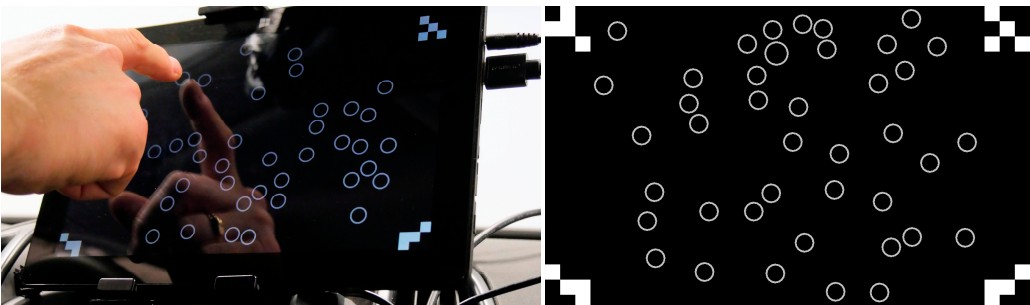

**Figure 7.** View of the screen used during SuRT task (source: Motor Transport Institute Warsaw; own picture).

## 5. Final Simulation Research Method of Driver Behavior at Pedestrian Crossings

The development of a final research method is contingent upon the implementation of a well-designed research plan and the delineation of a specific scope and research scenarios. This method should be validated and, if feasible, verified for the possibility of additional experiments. For instance, the analysis of alternative pedestrian crossing markings could be a potential avenue for further investigation.

### 5.1. Validation of Final Research Method of Driver Behavior at Pedestrian Crossings

The objective of the designed and planned simulation studies at this stage of the research is not to define the patterns of driver behavior, but rather to identify hazard factors (accident risk) in order to select the determinants of safety at pedestrian crossings.

The selection of a statistical measurement sample in each study that is to be used to define general conclusions based on individual measurements is a key assumption that confirms the validity of the defined general conclusions. This is particularly important when researching human behavior, which always has a large subjective and individual component. The aim of such research is often to classify and recognize typical behavior patterns. When analyzing the aforementioned problem, namely the perception of the traffic scene while crossing pedestrian crossings and the behavior of drivers as a result of various driving scenarios and the influence of vulnerable road users, it is of the utmost importance that each of the examined individuals undertake the same journey, in the same conditions, and at a repeatable time. To minimize the impact of dependent variables and ensure the repeatability of the research, assumptions were made of identical driving scenarios on

exactly the same road infrastructure under the same weather conditions. Additionally, before each test, the measuring devices were calibrated, which eliminated errors resulting from, for example, differences in visual performance during eye tracking tests. In light of the aforementioned considerations, consultations were held with employees of the Transport Psychology and Driving Simulators Laboratory of the Motor Transport Institute in Warsaw (Poland), who have extensive experience in conducting research using simulators and research on the human factor in transport. As a result, the number of 30 study participants was deemed to be a representative measure.

This study examines the context of driver behavior at pedestrian crossings with different signage systems and locations on road infrastructure. It considers the impact of distractions on drivers and the varying patterns of other vulnerable road users. In a study with an experimental scheme, 30 drivers are planned to participate, including 15 men and 15 women. The prerequisites for participation are possession of a cat B driver's license and active participation in traffic for at least half a year. The respondents will be divided into two research groups based on age (young people—under 45 years old, and older people—over 45 years old). Remuneration is expected for participation in the study. The experiment will be conducted according to the scheme shown in Figure 4. The study will be individual–dual in nature.

At this stage, in addition to verifying the ability of the recruited individuals to carry out the study, the verification of the correct operation of all control and measurement equipment should be carried out. If the study involves the use of eye tracking systems during the study, the eye tracking system should always be checked to ensure it is working correctly. The calibration process of the eye tracking device is depicted in Figure 8. The specific nature of the research resulting from, for example, the type of simulator and the context of the test requires carrying out tests and adaptive drives. This allows for a verification of whether the planned and designed tests enable the assessment of driver behavior for the adopted research contexts, the scope of recorded signals is sufficiently selected, and the recording devices are properly set and configured. Preparations for adaptive driving are shown in Figure 9.

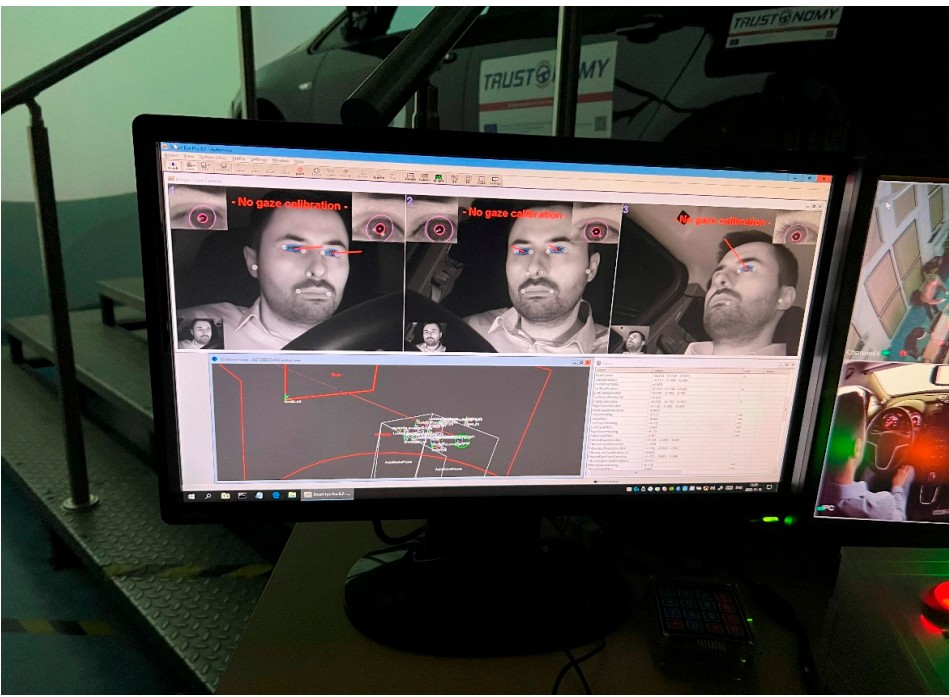

**Figure 8.** Calibration process of the eye tracking device (own picture).

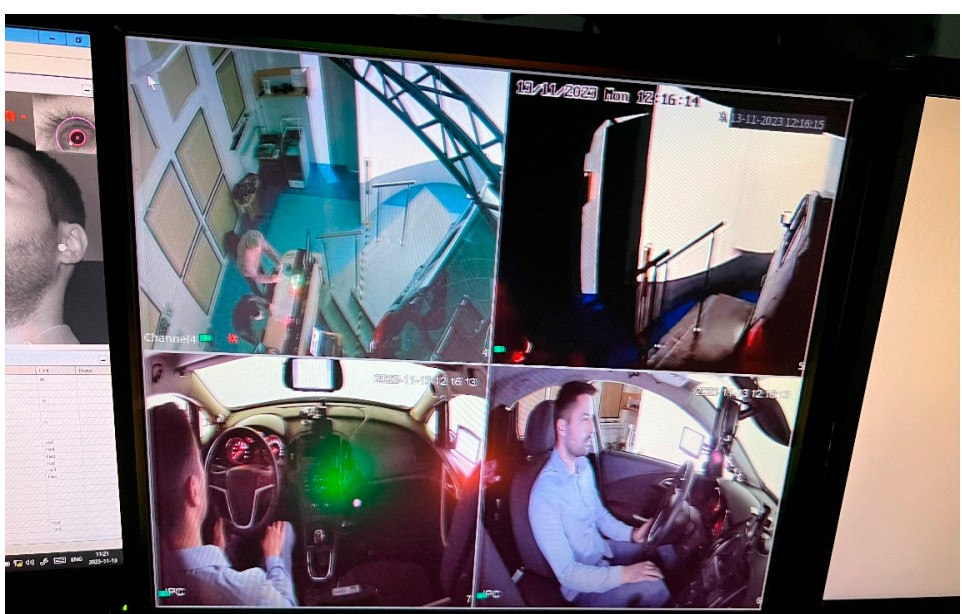

**Figure 9.** Preparations for the adaptation drive (own picture).

We decided to use a scenario in which the participants have to drive about 10 min of the route, which initially runs on a highway, then on a suburban road, and for a small section also with local buildings. The adaptation will make it possible to become familiar with controlling a simulated vehicle. It also aims at eliminating particularly sensitive individuals, who are experiencing symptoms of simulator sickness [32].

The adaptive driving will be followed by a research test, consisting of three drives and two training runs of a secondary task. The experiment is planned to include three test scenarios (three drives for each participant), including one run without distractors (in scenario one) and two runs in which the driver will perform additional tasks during the run: N-back or SuRT (in scenarios two and three) [41,42]. The main task in all three experimental runs is to drive along a specially designed section of the route with different types of pedestrian crossings and the presence of other vulnerable road users with normal and abnormal behavior patterns. Each scenario included areas with different landscape characteristics:

- Urban infrastructure;
- Suburban infrastructure;
- An area with sparse infrastructure;
- Forest area;
- An area without infrastructure.

In each of the above-mentioned areas, drivers encountered four pedestrian crossings. A total of twenty crosswalks were designed in each research scenario. Seven variants of signage were used to mark the locations of the pedestrian crossings, including a combination of current and experimental signages. The variants used are described below:

- Variant 1: sign D-6 "pedestrian crossing";
- Variant 2: sign D-6 "pedestrian crossing" and horizontal sign "airport" in the form of a permanent light signal;
- Variant 3: sign D-6 "pedestrian crossing" and horizontal sign "airport" in the form of a flashing light signal;
- Variant 4: sign D-6 "pedestrian crossing" and variable message sign (VMS) "STOP" in the form of a solid light signal;
- Variant 5: sign D-6 "pedestrian crossing" and variable message sign (VMS) "STOP" in the form of a flashing light signal;
- Variant 6: sign D-6 "pedestrian crossing" and variable content sign (VMS) "pictogram" in the form of a solid light signal;

- Variant 7: a D-6 "pedestrian crossing" sign and a variable content sign (VMS) "pictogram" in the form of a flashing light signal.

For half the length of each of the above-mentioned areas, a lane for cyclists will be designated on both sides of the roadway. Half of the crossings in each area will be accompanied by a bicycle crossing; so, in these areas, the D-6 "pedestrian crossing" sign will be appropriately replaced by the D-6b "pedestrian crossing and bicycle crossing" sign.

At the crossings, study participants will encounter six traffic situations involving vulnerable road users:

- A pedestrian walking normally;
- A pedestrian who, after entering the crossing, stops and turns around;
- A pedestrian who, after stepping into the crosswalk, stops for a moment and continues walking;
- A pedestrian running across a crosswalk;
- A pedestrian in an inebriated state (varying speed and angle of movement);
- A cyclist crossing a bicycle crossing.

In each scenario, vulnerable road users will always enter the pedestrian crossing from the right side. Randomizations of the order of appearance of vulnerable road users and the direction of travel were performed.

Two speed-adjusted layouts of the same road were designed, differing in the spacing between the pedestrian crossings. Test subjects at the beginning of each of the three experimental crossings will be asked to maintain speeds of 30, 50, 70, or 90 km/h. Each participant will take one slow route (at 30 or 50 km/h) and one fast route (at 70 or 90 km/h). For the third run, the subjects will be assigned pseudo-randomly, taking either the slow or fast route.

In scenarios 2 and 3, in addition to the main task, test drivers will be asked to perform an additional task. The experimental drive is designed as two variants (i.e., with two types of distractors: the N-back task or the SuRT task). The test subjects will be randomly assigned to one of two groups, each of which will drive through the scenario performing a different task first. The research drive with the additional task will be preceded by the task training.

The additional task is designed to induce cognitive load by engaging the driver's working (short-term) memory. In both tasks, less complex versions of the task are performed first, and then the level of difficulty is increased. The first task that we decided to use in this study was the delayed digit recall task, or auditory–vocal task, developed on the basis of the work of [32]. It involves listening to series of numbers and recalling the number heard: n (0–2) numbers backwards. This task has three levels of difficulty (0-back, 1-back, and 2-back). For the 0-back task, the participant must recall the last number heard. For the 1-back task, the participant recalls the number before the last number heard out of the 10 in series. The last version of the task is the 2-back task. Here, each time the speaker reads a list containing 10 numbers, the subject's task is to repeat out loud the number that was said third from the end. For example, if the digit 7 was spoken, then 4, and finally 5, the subject should say only the digit 7. For the purposes of this study, we decided to use the 1-back version of the task [43].

The second additional task to be used in this experiment was the SuRT task. This is a visual–manual task [43]. It involves searching a touch device for the correct circle (the largest one) among a dozen circles appearing on a screen. For the SuRT task, a touchscreen tablet placed in the central part of the driver's desktop will be used. The location of the tablet is determined so that it is in the peripheral field of vision, outside of the driver's acute vision [43].

At the end of the study, a separate questionnaire will be used to directly evaluate the experimental signs implemented in the scenarios.

## 5.2. Alternative Markings for Pedestrian Crossings without Traffic Lights

In addition to the use of the conventional method of marking pedestrian crossings and the use of conventional traffic signs informing the driver of the situation in which he or she is in the vicinity of a pedestrian crossing, it is also possible to use internally developed alternative signs. During the execution of the crossing procedure according to the selected driving scenario, three types of alternative signage can be implemented. Two of these can be mounted in a manner analogous to conventional vertical signs, where VMS variable content signs can be used for their implementation. The aforementioned signs should be presented to the study participants in a variety of ways, either as fixed signage or with a variable frequency of appearance of information on the variable content sign. In this case, an inscription and pictogram could be used. Furthermore, the possibility of implementing an alternative sign directly on the roadway should be considered. This would allow for the direct and immediate notification of drivers approaching a pedestrian crossing, without the need for signaling, depending on the distance at which the driver was. The conceptual graphics of each solution are presented below (Figures 10–12).

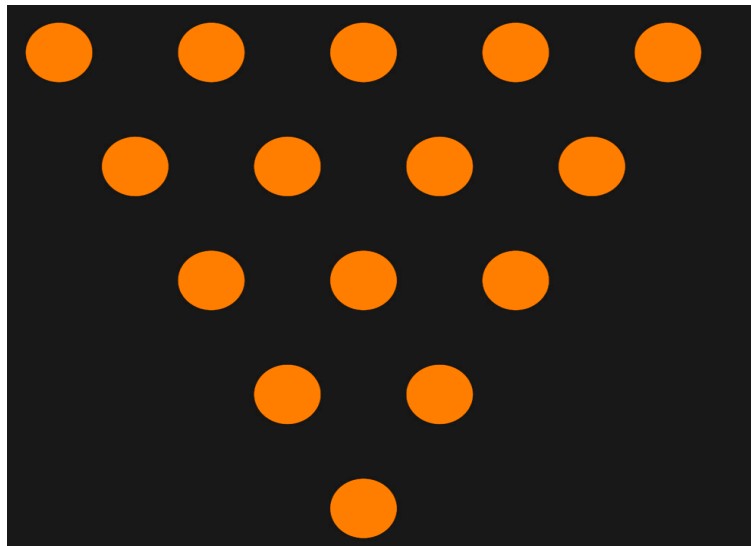

**Figure 10.** Alternative sign "airport" implemented in the transit scenario.

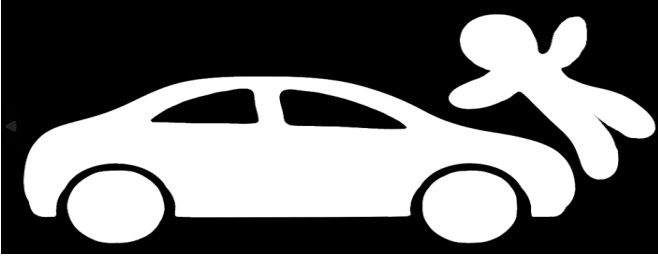

**Figure 11.** Alternative sign "pictogram" implemented in the transit scenario.

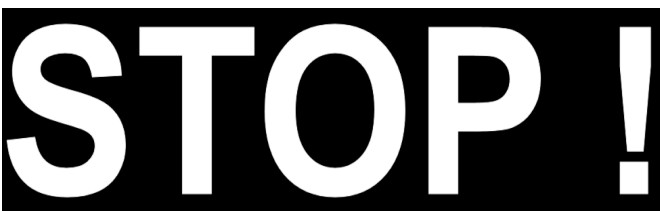

**Figure 12.** Alternative sign "STOP" implemented in the transit scenario.

*5.3. Additional Surveyed Questions*

One of the additional research problems may be the impact of a variable message sign with appropriate pictograms on the perception and reaction of drivers. In this scenario, it is important to obtain additional information regarding the perception of such signs and their subjective assessment in the context of approaching and crossing a pedestrian crossing. One of the assumed utilitarian goals of the planned research is the selection of an innovative safety support system at pedestrian crossings [44]. One of the elements of such a system may be an appropriate variable message sign.

During the designed study, in addition to the basic metric required during simulator research related to health predisposition for this type of research, a separate one can be added concerning the direct evaluation of sign implementation [45]. This survey could be used to compare the results of simulator research with the direct evaluation of research participants. Below are the sample questions planned for use:

✓ In your opinion, was the pictogram or the STOP sign more legible during the study?
✓ Which of the pictograms and text signs presented is the most understandable to you?
✓ Were there any elements along the route that were confusing to you in some way during the ride? If "yes", please specify which ones, if "no" please skip the question.
✓ In your opinion, did the signs that appeared guarantee sufficient time to react in a situation where a pedestrian appeared at a pedestrian crossing? If "no" please justify why, if "yes" please skip the question.
✓ In your opinion, what did the light signs built into the roadway mean?
✓ Is the sensation of driving in the simulator in your case comparable to driving a conventional vehicle in normal traffic?
✓ In your opinion, is the placement of variable message signs (pictograms and text signs) on the screen on the right side of the road optimal and does not cause discomfort when reading the intent of the sign? If "no" please justify why, if "yes" please skip the question.
✓ Which of the signs was visually "better" for you to perceive?
✓ Which of the signs was more difficult for you to understand and required more attention?
✓ Which of the signs shown do you think could be used in front of a pedestrian crossing? (you can mark more than one answer)
✓ Were any of the behaviors of pedestrians or cyclists surprising to you? If "yes" please justify which ones, if "no" please skip the question.
✓ How long have you held your driver's license?
✓ How many kilometers on average do you drive per year as a driver?
✓ Are you a professional driver?
✓ Have you been fined for speeding in the last 5 years?
✓ Do you have a visual impairment?

## 6. Conclusions

This study presents the methodology for designing and planning research on driver behavior at pedestrian crossings using whole-vehicle simulators, taking into account the context. Therefore, a flowchart of the research planning process, decomposition of tasks using WBS, and estimating time-consuming implementation, as well as an example implementation time schedule, are presented.

It was determined that dedicated research should be conducted in specific problem contexts. The problems identified were the identification of hazards and the risk of accidents involving vulnerable road users. The purpose of this identification is to determine the determinants of safety at pedestrian crossings, which should constitute guidance when designing new solutions for safety support systems at pedestrian crossings. A number of hazard factors were identified, divided into categories including types of crossings, location and surroundings, and behavior of vulnerable road users and attention (focus) distractors, both inside and outside the vehicle. A method for defining and selecting research scenarios and selecting a group of research participants was proposed. Additionally, it was

proposed to conduct repeatable test scenarios for different driving speeds and different weather conditions. With respect to the publications on this topic, this work focuses on the process of designing and planning dedicated simulation studies, which may provide a source of guidance and good practices for other researchers. This is an example of how interdisciplinary research involving human factors, traffic organization, and ITS systems can be planned and implemented.

The analysis of collected data on direct driver behavior and vehicle movement parameters should follow the simulation study. This includes, but is not limited to, the vehicle's instantaneous speed, the degree of brake lever pressure, and the number of collisions. It is also important to check the correctness and accuracy of each task's execution, including the number of mistakes in the SuRT and N-back tasks. It is important to include information on the vehicle's position in the lane, including its position in the lane cross-section and the standard deviation of the position. Additionally, it is important to collect information on minor steering wheel corrections. All these parameters can be monitored and recorded during tests carried out on the indicated AS 1200-6 passenger car simulator.

This study presents additional surveys that could be included. Once all trials are completed, participants can be asked to fill out a questionnaire. However, it is important to note that the validation of the use of driving simulators to conduct research on driver safety and behavior should be based on objective evaluations. Conducting in situ tests carries a significant risk of causing accidents. However, it is important to validate the correctness of the results obtained under controlled and safe real-world conditions.

**Author Contributions:** Conceptualization, R.B., D.S., M.K., A.N., A.B.Z., J.M. and E.D.; Methodology, R.B., D.S., M.K. and A.N.; Software, M.K. and A.N.; Validation, R.B.; Investigation, R.B. and D.S.; Resources, M.K., A.N. and K.G.; Writing—original draft, D.S.; Writing—review & editing, R.B., D.S., M.K., A.N., K.G., A.B.Z., J.M. and E.D.; Supervision, R.B.; Project administration, R.B.; Funding acquisition, R.B. and D.S.; Data curation, A.N. All authors have read and agreed to the published version of the manuscript.

**Funding:** This research was funded by Polish Ministry of Education and Science grant number 32/014/SDW/005-52 (DWD/5/0590/2021).

**Institutional Review Board Statement:** In Polish law, the ethical committee opinion is mandatory only regarding medical experiments (according to Act of 5 December 1996 on the profession of doctor and dentist). For the research presented in the article the procedure doesn't apply. In 2016, Polish National Science Centre provided recommendations regarding research involving human participation (other than medical) which was followed in the research. Motor Transport Institute provided for each participant the appropriate set of consents and descriptions including consent to participate in the study, GDPR consent, experiment description, the list of contraindications and description of possible influence of simulator for the participant. During the experiment the level of influence of the simulator sickness was also monitored by questionnaires. All participants were informed about possibility to resign from the experiment at any time without giving justification.

**Informed Consent Statement:** Informed consent was obtained from all subjects involved in the study.

**Data Availability Statement:** The original contributions presented in the study are included in the article, further inquiries can be directed to the corresponding author.

**Conflicts of Interest:** The authors declare no conflict of interest.

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
