# Peer review of "Designing and Planning of Studies of Driver Behavior at Pedestrian Crossings Using Whole-Vehicle Simulators"

_applsci, doi:10.3390/app14104217_

Round 1

Reviewer 1 Report

Comments and Suggestions for Authors

 Overview

This is a potentially interesting methodology paper on driving simulation, but due to very unclear writing and a lack of key details on designing and conducting simulation experiments, it falls short. Key simulation methodology details—like what measures a simulator can provide, what simulators (especially affordable simulators) are available, how to design scenarios, and what are key issues in validation are absent, which unfortunately lead me to reject this article for publication.

Specific comments:

Abstract (and elsewhere): “Disease” is not an appropriate synonym for “simulator sickness”—just using “sickness” or “discomfort” is probably better.

Line 234: At this point, the research problem described here, “the behavior of drivers around different pedestrian crossings with different factors influencing drivers’ perception and focus” is very vague, and it isn’t clear what the problem is that the researchers are addressing—accidents between vehicles and vulnerable road users at intersections, potentially caused by driver inattention or misallocated attention? A “problem” should be an undesirable phenomenon that the researchers are addressing in order to solve, and it would be useful to readers to describe it in more detail. For example, I’m unsure if the factors such as “large-area advertising” is being investigated as a possible distractor for drivers or for pedestrians. 

Line 271: What does “ba-test” mean? 

Line 284: Motion platform driving simulators in real vehicle cabins, especially with six degrees of freedom, are relatively rare and likely only available to researchers who are already very familiar with the topics covered in this paper. Can you describe less expensive alternatives, such as driving bucks, desktop simulators, or even VR (which admittedly has simulator sickness issues, as you note, but is potentially an affordable option for smaller research labs)?

Line 290: Whether or not the key variables of interest are repeated measures (i.e., a within-subjects comparison) or not is an important factor determining the sample size, as is the effect size of the expected outcome variable—I think these should be discussed here, as many driving studies can use fewer subjects than 30, while others require many more than 30 (especially if a scenario cannot be reasonable repeated for a subject, and a between-subjects approach is required).

Figure 4: It’s hard to read any text on this figure, and given the steps are more clearly outlined in the following table, I would consider eliminating it.

Line 403: “Dependent variable” or “variables” should be included somewhere here—that’s what most readers are going to associate with “driver reactions,” which I believe is the phrase you’re using to indicate “driver behaviors to be recorded and analyzed.” There are many such behaviors / variables, and they aren’t really discussed here—from vehicle speed, headway, lane position and variability, number of violations or crashes, to attention-related variables (total off-road glance time, mean glance duration to specific locations), to scenario-specific measures (reaction time to a red light turning greed; path through an intersection; distance from pedestrians in crosswalk, etc.). The analysis part, where you get into categorization via neural networks, is probably out of scope for a paper like this; there should be instead a focus on the kinds of measures that can reasonably be collected during simulator work, along with some of the challenges (operationalization; dealing with big data that is often exported from simulators, along with eye trackers).

460: What is “ba-research,” which appears twice in this paragraph?

Figure 7: This image shows a part of the calibration process for eye-tracking, not simulation.

Comments on the Quality of English Language

There are many sentences that are difficult to understand, including sentence fragments in the Abstract. Because it is a methodology paper, and not reporting scientific findings, there are not many tables or graphs to help make sense of the key points, and thus is very challenging to read and understand.

Author Response

Thank you for your comments. We have gone through your comments carefully and tried our best to address them one by one. We hope the manuscript has been improved accordingly. We have made changes in the manuscript in response to comments, and highlight the changes in the manuscript.

We appreciate you and the reviewers for your precious time in reviewing our manuscript and providing valuable comments. It was your valuable and insightful comments that led to possible improvements in the current version. Attached we provided the point-by-point responses. All modifications in the manuscript have been highlighted in yellow. 

Reviewer 2 Report

Comments and Suggestions for Authors

Dear authors,

Your study has merit to a certain extent. The topic is relevant, but the study purposes, results and conclusions are quite elusive. The problem lies in the way the work is presented and defended, that does not make justice to underlying research work. The text is very long and repetitive, and uses a generic and sometimes circular language. One cannot present a scientific study without clearly demonstrating its added value, compared to other published works. That said, the following aspects should be addressed before publication:

1) The abstract must be rewritten. The first two thirds are entirely general, and sometimes no more than an expression of common sense (in this area of research). The reference to your study appears only in the penultimate sentence of the abstract. But its purpose and meaning are unclear: «This article also describes the dis- ease [disease?] of using simulators to conduct research activities, along with the workflow from assumptions to finalizing the research.»

What research activities (nonspecific language)? Is the purpose of the study to provide a timeline of the research conducted or a comprehensive regulatory protocol?

When the authors state « An algorithm is also presented for the study of individual participants, with examples for separate research scenarios concerning driver behavior in the vicinity of pedestrian crossings.», once again the language is nonspecific and this is the last sentence. Specifically, what do the authors intend to study about individual participants and in what scenarios?

The abstract is not concise, the contextualization of the study is poor, its objectives are not clear, its novelty is not presented and no allusion is made to results and conclusions.

2) The introduction to section 2 is written almost as simple guidelines on how to write an article and, in particular, how to carry out a literature review. Its relevance in the context of an article is nil.

3) In section 4.1., «Defining the research context and assumptions», the assumptions are unclear. There seems to be confusion between objectives, analysis of results and assumptions. This section should be clarified.

4) In section 4.2., «Selection of the type of the driving simulator», the language is redundant (repetitive) and generic. For instance, when the authors say «The research parameters in the case of driving simulator research may be driving speed (…)», are you saying that it may be a parameter to consider or that it has been considered? Once again, the language is unassertive.

5) In section 4.3., «Research planning», the purpose of the planning is elusive. It is presented as a schedule of tasks for a research project or thesis. What is the point of knowing the time allocated or available for carrying out the different tasks?

Note that section 4 precedes the conclusion chapter, thus it should present results. But is it relevant for carrying out a proper simulation and establishing a useful protocol (for others) to list these tasks and their duration? Tasks such as «software testing», «corrections and improvements» or «development of the research report». This could apply to a multitude of research projects or thesis planning.

Are the durations presented part of the planning that has been defined as the objective of the study? If so, it would be absurd. Please clarify.

6) The fact is that the manuscript is not correctly structured: there is no methods and materials section, which is scattered throughout the text; the results of the study are ambiguous and the relevance of the research workflow (protocol?) is unclear and not demonstrated at all. To this end, and considering that simulator sickness is a relevant research topic, the participants' behaviors and opinions should have been quantitatively typified through an appropriate survey and quantified in relation to the measured parameters of the simulations carried out (as alluded). In addition, a comparison should be made with other studies to establish its relevance.

7) Furthermore, there is a total lack of critical analysis of the results and the conclusions are innocuous. Some of the considerations made are almost an expression of common sense. The citation of other studies (for the first time) in this context, as well as the use of generic language and the use of purely qualitative considerations, without it even being clear whether these are the authors' own conclusions, undermines the relevance of the study and does not do it justice.

The authors should review the structure of the article.

These aspects should be properly addressed and reviewed. Perhaps the authors should consider submitting this study to a journal in the field of transport or accident analysis and modeling.

The work has merit, but the authors must improve the manuscript before publication.

Best regards,

Comments on the Quality of English Language

Moderate editing of English language required.

Author Response

(The authors gave the same response as above.)

Reviewer 3 Report

Comments and Suggestions for Authors

This study explores the utilization of driving simulators as a research tool, detailing their effectiveness in replicating real-world conditions and facilitating research on various aspects of human behavior, particularly focusing on driver behavior near pedestrian crossings. It’s meaningful and interesting. Comments are as follows:

i). Please provide a flowchart to enhance readers' understanding of the proposed simulation testing methods in this study.

ii). The participants in this study represent to some extent a broader population, but specific information regarding sampling methods and participant characteristics requires further data and analysis to determine.

iii). It is recommended to supplement data analysis to validate the accuracy of the conclusions drawn in this study.

iv). Please provide the design rationale for the Additional Surveyed Questions.

v). Please carefully check the grammar and reference format.

Comments on the Quality of English Language

The language needs more work.

Author Response

(The authors gave the same response as above.)

Reviewer 4 Report

Comments and Suggestions for Authors

 Title: Planning studies of driver behavior at pedestrian crossings using whole-vehicle simulators

This manuscript "Planning studies of driver behavior at pedestrian crossings using whole-vehicle simulators" is a comprehensive study that presents the complete process involved in conducting simulator studies, from general examples defining the framework related to the simulator study approach, to ready-to-use guidelines with study scenarios. Due to the specific nature of simulator testing and objections related to long-term testing and a representative sample (number of subjects and scenarios), the authors presented in this paper a comprehensive method and algorithm for planning and designing simulator studies.

1.  I think the article is too long. The paper writes about things that are implied when doing research, then repeats similar material or identical parts of the text several times. It is important to note that the goal of this paper is not to write instructions on how to write a scientific paper or conduct research, and often, reading this article, one gets the feeling that the authors are writing instructions.

Below are some examples.

a.      In the text in chapter 2, it is explained that it is important to research, that is, to review and analyze a lot of papers, what other authors have written about a similar problem so that research is not carried out unnecessarily. -  for a paper of this nature it is not necessary to write, that goes without saying.

b.      ’The research problem and context determine the entire planning, design, and implementation processes. The correct and precise definition of the research context and the definition of research problems or hypotheses determine the validity of the selection of methods and the scope of the research. …” - it is not necessary to write, that goes without saying.

c.      Task steps listed in the Gantt chart are reprinted below the Gantt chart. – unnecessary

d.     Lines 460-466: Repetition of previously written text.

e.      Lines 490-499... are the same as lines 290-299. It is also interesting to note that the authors refer to different literary sources for the same text.

.

.

.

2.     2.  None of the figures presented in the paper are mentioned anywhere in the text. The only figure the authors refer to in the text is Figure 1, but what they are referring to is not found in Figure 1.

3.     3. I suggest that the authors create Figure 2 themselves, which refers to the types of distractors.

4.     4.  The space between lines in the enumeration (eg 247 to 260) is unnecessarily large. The words on the Gantt chart should only be in English.

5.      5. In the chapter Defining the research scenario, it is stated that to research the behavior of drivers, the behavior of a group of 900 drivers approaching a pedestrian crossing was analyzed. From what has been written, it is not clear to the reader whether the research was carried out by the authors of this paper, or whether the authors are describing in detail someone else's research, given that they refer to literary source 30 as part of this presentation. I'm also not sure that the literature source 30 is an appropriate source at this point.

6.     6.  Figures 5 and 6 have the same name.

7.     7.  The conclusion needs to be rewritten. This conclusion consists of a set of unrelated sentences. It is also not clear why the authors refer to literary sources in the conclusion. The conclusion cannot be written as an instruction.

 8.The paper should be written in such a way that it is clear to the reader what the authors have done and what they have taken from other sources and research. It is often unclear in this work. It is also necessary to check throughout the paper whether the cited sources are appropriate for the part of the text behind which they are cited.

Author Response

(The authors gave the same response as above.)

Reviewer 5 Report

Comments and Suggestions for Authors

This paper describes a method to perform simulator studies, particularly simulator studies regarding driver behaviour at pedestrian crossings.

The way you explain this method is quite clear. However, the method focusses very much on the simulation of driving behaviour near pedestrian crossings. This makes the method not very general to apply. So you could better shorten the description of this method and add some results of applying this simulation method.

Author Response

(The authors gave the same response as above.)

Round 2

Reviewer 2 Report

Comments and Suggestions for Authors

Dear authors, 

The manuscript has been improved.

Best regards, 

Comments on the Quality of English Language

Moderate editing required. 

Author Response

Dear Reviewer,

We're so grateful to you for taking the time to read our manuscript and for all your thoughtful comments and suggestions. They've been invaluable in helping us to improve and revise our work.

We're also really thankful for your kind words about our efforts in revising and improving the manuscript and for your final acceptance and recommendation for publication.

With warm regards,

Authors

Reviewer 3 Report

Comments and Suggestions for Authors

All the reviewer comments have been addressed, and the paper is recommended for publication.

Author Response

(The authors gave the same response as above.)

Reviewer 4 Report

Comments and Suggestions for Authors

Last time I commented size of the paper, considering that the work was unnecessarily long due to the repetition of descriptions of the same things. I don't see that a contribution was made in terms of formatting the description, only some new paragraphs were added.

In the first chapter, part of the text was added (lines 105-112). I assume that one of the reviewers was looking for additional literary sources. However, it was done in the wrong way. After the reference to literary source 19, the reference to literary source 41 follows. It is necessary to renumber the literary sources. The same mistake was made in chapter 4.2 (lines 300-313)

It is not necessary to write ''own work'' after the name of figures. It is understood that the authors created a figure, graph, or table themselves if they did not refer to a source.

Last time I commented that it is not necessary to show the same things in several ways. This referred to Figure 4 and the listing below Figure 4 (paper version 1). In the new version of the paper, one of the views was not removed, but a flow diagram was added that shows the same thing. So now we have the same thing shown in three ways.

Author Response

Dear Reviewer
we would like to thank you very much for all your comments and suggestions, which have significantly helped to improve and revise our manuscript.
In response to the suggestions made:
As per your suggestion, the references and citations in the body of the manuscript have been corrected and renumbered.
Removed unnecessary wording ‘’own work‘’ after the name of figures.

Regarding the suggestions related to the length of the manuscript and the similar content shown in Figures 4 and 5 and Table 1, we would like to clarify that these additions are due to the suggestions of the other reviewers. We have decided to leave these sections of the article as well as the article as a whole unchanged, as we have received positive feedback on the revised manuscript from the other four reviewers and satisfaction with the corrections made in the manuscript.

Kind Regards

Authors

Reviewer 5 Report

Comments and Suggestions for Authors

Thank you for improving your paper. You have given a extensive description of how to plan these kind of simulator studies. The level of the description is relevant for the type of studies you have in mind.

Author Response

(The authors gave the same response as above.)
